# Socioeconomic level and associations between heat exposure and all-cause and cause-specific hospitalization in 1,814 Brazilian cities: A nationwide case-crossover study

Rongbin Xu[1,2], Qi Zhao[2], Micheline S. Z. S. Coelho[3], Paulo H. N. Saldiva[3], Michael J. Abramson[2], Shanshan Li[2]*, Yuming Guo[1,2]*

1 School of Public Health and Management, Binzhou Medical University, Yantai, China, 2 School of Public Health and Preventive Medicine, Monash University, Melbourne, Victoria, Australia, 3 Institute of Advanced Studies, University of São Paulo, São Paulo, Brazil

* shanshan.li@monash.edu (SL); yuming.guo@monash.edu (YG)

**Data Availability Statement:** The authors are not permitted to share the data owned by a third party.

## Abstract

### Background

Heat exposure, which will increase with global warming, has been linked to increased risk of a range of types of cause-specific hospitalizations. However, little is known about socioeconomic disparities in vulnerability to heat. We aimed to evaluate whether there were socioeconomic disparities in vulnerability to heat-related all-cause and cause-specific hospitalization among Brazilian cities.

### Methods and findings

We collected daily hospitalization and weather data in the hot season (city-specific 4 adjacent hottest months each year) during 2000–2015 from 1,814 Brazilian cities covering 78.4% of the Brazilian population. A time-stratified case-crossover design modeled by quasi-Poisson regression and a distributed lag model was used to estimate city-specific heat–hospitalization association. Then meta-analysis was used to synthesize city-specific estimates according to different socioeconomic quartiles or levels. We included 49 million hospitalizations (58.5% female; median [interquartile range] age: 33.3 [19.8–55.7] years). For cities of lower middle income (LMI), upper middle income (UMI), and high income (HI) according to the World Bank's classification, every 5°C increase in daily mean temperature during the hot season was associated with a 5.1% (95% CI 4.4%–5.7%, $P < 0.001$), 3.7% (3.3%–4.0%, $P < 0.001$), and 2.6% (1.7%–3.4%, $P < 0.001$) increase in all-cause hospitalization, respectively. The inter-city socioeconomic disparities in the association were strongest for children and adolescents (0–19 years) (increased all-cause hospitalization risk with every 5°C increase [95% CI]: 9.9% [8.7%–11.1%], $P < 0.001$, in LMI cities versus 5.2% [4.1%–6.3%], $P < 0.001$, in HI cities). The disparities were particularly evident for hospitalization due to certain diseases, including ischemic heart disease (increase in cause-specific

For information on data access, readers are asked to contact Dr. Pei Yu (Pei.Yu1@monash.edu).

**Funding:** R.X. was supported by China Scholarship Council (201806010405). S.L. was supported by an Early Career Fellowship of the Australian National Health and Medical Research Council (APP1109193). Y.G. was supported by a Career Development Fellowship of the Australian National Health and Medical Research Council (APP1107107 & APP1163693). The funders had no role in study design, data collection and analysis, decision to publish, or preparation of the manuscript.

**Competing interests:** I have read the journal's policy and the authors of this manuscript have the following competing interests: MA holds investigator initiated grants from Pfizer and Boehringer-Ingelheim for unrelated research and an unrelated consultancy from Sanofi. He has also received assistance with conference attendance from Sanofi and a speaker's fee from GSK. YG is an Academic Editor on PLOS Medicine's editorial board. The other authors declare no actual or potential competing interests.

**Abbreviations:** COPD, chronic obstructive pulmonary disease; GDP, gross domestic product; HI, high income; LMI, lower middle income; RR, relative risk; SES, socioeconomic status; UHI, urban heat island; UMI, upper middle income.

hospitalization risk with every 5°C increase [95% CI]: 5.6% [−0.2% to 11.8%], $P = 0.060$, in LMI cities versus 0.5% [−2.1% to 3.1%], $P = 0.717$, in HI cities), asthma (3.7% [0.3%–7.1%], $P = 0.031$, versus −6.4% [−12.1% to −0.3%], $P = 0.041$), pneumonia (8.0% [5.6%–10.4%], $P < 0.001$, versus 3.8% [1.1%–6.5%], $P = 0.005$), renal diseases (9.6% [6.2%–13.1%], $P < 0.001$, versus 4.9% [1.8%–8.0%], $P = 0.002$), mental health conditions (17.2% [8.4%–26.8%], $P < 0.001$, versus 5.5% [−1.4% to 13.0%], $P = 0.121$), and neoplasms (3.1% [0.7%–5.5%], $P = 0.011$, versus −0.1% [−2.1% to 2.0%], $P = 0.939$). The disparities were similar when stratifying the cities by other socioeconomic indicators (urbanization rate, literacy rate, and household income). The main limitations were lack of data on personal exposure to temperature, and that our city-level analysis did not assess intra-city or individual-level socioeconomic disparities and could not exclude confounding effects of some unmeasured variables.

## Conclusions

Less developed cities displayed stronger associations between heat exposure and all-cause hospitalizations and certain types of cause-specific hospitalizations in Brazil. This may exacerbate the existing geographical health and socioeconomic inequalities under a changing climate.

## Author summary

### Why was this study done?

- Heat exposure is associated with increases in mortality and morbidity, but vulnerability to heat is not evenly distributed.

- It remains uncertain whether the magnitude of heat impacts on all-cause and cause-specific hospitalizations is associated with local socioeconomic characteristics.

### What did the researchers do and find?

- We performed case-crossover analyses based on a nationwide dataset with 49 million hospitalizations in the hot season during 2000–2015 in 1,814 Brazilian cities.

- We found that less developed cities showed higher increased all-cause and cause-specific hospitalization than more developed cities when exposed to the same temperature rise during the hot season.

- These inter-city socioeconomic disparities in associations of heat with health outcomes were particularly notable for hospitalizations of children and adolescents, and for hospitalizations due to ischemic heart disease, asthma, pneumonia, renal diseases, mental health conditions, and neoplasms.

## What do these findings mean?

- Increasing heat exposure along with global warming could be a potential driver for exacerbating inter-city health inequalities.

- More attention should be paid to less developed cities in Brazil to tackle the morbidity burden of heat exposure, in order to promote health equity under global warming.

## Introduction

Climate change is a major health threat in the 21st century via many pathways, including causing more heat-related mortality or morbidity [1]. Globally, although people living in low- and middle-income countries bear less responsibility for the global warming, they are actually more affected by temperature rise than residents in high-income countries [1]. Climate change has been viewed as a potential driver for exacerbating inequality. However, the modification effect of regional socioeconomic status (SES) on the heat–mortality association remains uncertain [2]. Some studies found that cities or areas with lower SES (e.g., lower urbanization [proportion of urban people], household income, education, or gross domestic product [GDP] per capita) tended to show stronger heat–mortality associations [3–9]. These associations mainly mirrored the residents' lower adaptation capacity and higher sensitivity to heat exposure [3–9]. In contrast, other studies found that more urbanized or developed areas showed stronger heat–mortality associations [10,11]. They attributed this phenomenon to the urban heat island (UHI) effect [10,11], i.e., that people living in urbanized settings tend to experience higher temperatures due to the great thermal-storage capacity of heavily engineered environments, poor ventilation, and localized heat sources (e.g., vehicles, air conditioners) [12]. There are also some studies that found no significant modification effect of local SES on heat–mortality associations [13–16].

In addition to the conflicting findings, some limitations of previous studies should be noticed. First, the number of locations was often limited (the largest study included 340 cities [10]), which may restrict the statistical power of these studies. Second, most studies were conducted in developed countries; evidence from developing countries was relatively rare. Third, most studies only focused on heat–mortality associations [3–11,13–16], while few of them addressed inequality in heat–morbidity (e.g., hospitalization, emergency department visit) associations. The existing heat–morbidity studies also showed inconsistent findings with respect to the modification effect of regional SES [17–20], and they also had a limited number of locations (up to 158 locations [18]). Finally, most previous studies only focused on all-cause or cardiovascular mortality or morbidity; little is known about inequality in other specific potentially heat-related conditions (e.g., renal diseases, diabetes, mental illness) [21].

In this study, we aimed to evaluate whether city-level heat–hospitalization (both all-cause and cause-specific) associations are modified by local socioeconomic level (literacy rate, urbanization, average household income, and GDP per capita), using a national hospitalization dataset from Brazil during 2000–2015.

## Methods

This time-stratified case-crossover study is reported following the Strengthening the Reporting of Observational Studies in Epidemiology (STROBE) Statement (see S1 STROBE Checklist).

We performed all the data analyses according to a prospective analysis plan (S1 Text). Following feedback during peer review, we made some modifications to the analysis plan, as described in S1 Text.

## Data collection

The hospitalization data and meteorological data sources have been described in detail in our previous publications [22–24]. Briefly, we collected all-cause hospitalization data from the Brazilian Unified Health System between 1 January 2000 and 31 December 2015. We included 1,814 cities located in 5 regions (north, northeast, central west, southeast, and south) with complete data for the entire 16 years, which covered 78.4% of the Brazilian population. In this study, a "city" (also known as a "municipality" in Brazil) refers to an administrative area with an officially defined boundary under the level of Brazilian states. Each city covers all urban and rural areas in its administrative area. The 16-year average population size of the 1,814 cities during 2000–2015 ranged from 8,014 to 11,048,690, with a median (IQR) of 29,814 (19,859 to 57,484). Generally, the population size of most cities (1,577 cities, 87.0%) was less than 100,000 (Fig A in S2 Text). The hospitalization dataset recorded sex, age, date of each admission, and the code of primary diagnosis according to the 10th revision of the International Classification of Diseases (ICD-10). According to previous reviews on heat and morbidity [21], we selected hospitalizations due to 16 specific causes, including all cardiovascular diseases, ischemic heart disease, heart failure, heart rhythm disturbances, cerebrovascular diseases, stroke, peripheral vascular disease, all respiratory diseases, chronic obstructive pulmonary disease (COPD), asthma, pneumonia, diabetes, renal diseases, mental health conditions, neoplasms, and heat illness (see their ICD-10 codes in Table A in S2 Text). This study was approved by the Monash University Human Research Ethics Committee, and individual consent was exempted because we used anonymized data.

City-specific daily mean temperatures were sourced from a 0.25˚ × 0.25˚ Brazilian meteorological dataset [25]. In order to estimate the association between hospitalization and heat exposure, we restricted our analyses to the hottest 4 consecutive months (hot season) for each city each year during the study period [26]. For each city, the hottest 4 consecutive months (e.g., November, December, January, February) were selected according to the average daily mean temperature of each specific month during 2000–2015 (e.g., the average temperature of all Januaries during the 16 years). With this method, the months of hot season held constant for a specific city during these years.

City-level socioeconomic levels were represented by 16-year (2000–2015) average literacy rate of people aged 15 years or above, 16-year average urbanization rate (proportion of urban population), average monthly household income per capita in 2010, and 16-year average GDP per capita. City-level literacy rate, urbanization rate, and population size came from the Brazilian Census 2000 and 2010; the data gaps for other years were filled by linear interpolation. City-level household income measures from the Brazilian Census 2000 and 2010 were not comparable due to different statistical criteria. We chose the latter because 2010 was in the middle of our study period; thus, the 2010 value tended to be a better surrogate of the 16-year average than the 2000 value. City-level GDP per capita for every year during 2000–2015 was reported by the Brazilian Institute of Geography and Statistics (BIGS). We adjusted all GDP per capita and household income data to 2015 US dollars, according to the Consumer Price Index during 2001–2015 and the average exchange rate in 2015. From the Brazilian Census 2010, we also collected the percentages of young population (0–19 years) and elderly population (60 years or above), as an indicator of population age structure. All these socioeconomic and demographic data were downloaded from the BIGS official website [23]. We then

stratified the 1,814 cities into 4 groups (Q1–Q4) according to the quartiles (from the lowest to the highest quartile) of each socioeconomic indicator. We also classified the cities into lower middle income (LMI, GDP per capita: US$1,146–US$4,035), upper middle income (UMI, US$4,036–US$12,475), and high income (HI, >US$12,475) according to the World Bank's 2015 standard. Because there was only 1 city with GDP per capita less than US$1,146 (US$978), we simply classified it as LMI.

## Statistical analysis

We used a 2-stage analysis to quantify the associations between heat exposure and risk of all-cause and cause-specific hospitalization [23]. In the first stage, we used a time-stratified case-crossover design to evaluated the heat–hospitalization association for each city [27]. Quasi-Poisson regression with a distributed lag model was used to estimate the city-specific association with the equation below [23]:

$$\text{Log}(Y_{it}) = \alpha + \text{cb}(\text{Temp}_{it}) + \beta\text{Strata}_{it} + \gamma\text{DOW}_{it} + \delta\text{Holiday}_{it} + \varepsilon_{it} \tag{1}$$

where $Y_{it}$ represented the daily counts of hospitalization in city $i$ on day $t$; $\alpha$ was the intercept; $\beta$, $\gamma$, and $\delta$ were the coefficients; and $\varepsilon_{it}$ was the residual error. $\text{Strata}_{it}$ was a stratum variable combining year and calendar month to adjust for long-term trend and inter-month seasonal variations. $\text{DOW}_{it}$ was a categorical variable to adjust for hospitalization variation within 1 week. $\text{Holiday}_{it}$ was a binary variable (public holiday or not) to adjust for potential impacts of public holidays on hospitalization. $\text{cb}(\text{Temp}_{it})$ was a 2-dimensional (exposure–response dimension and lag–response dimension) cross-basis function to model the lagged associations of daily mean temperature with hospitalizations. In each dimension, a specific smoothing function (e.g., linear function, natural cubic spline) can be used to define the shape of the relationship [26,28]. According to our preliminary analyses described before [23], we used a linear function for the temperature–response dimension, and a natural cubic spline with 3 degrees of freedom for the lag–response dimension along 0–7 lag days.

In the second stage, we pooled the city-specific estimates for all cities or cities in different socioeconomic groups (Q1–Q4 of each socioeconomic indicator, or classification according to the World Bank), using a random effect meta-analysis with maximum likelihood estimation [29]. This provided us a pooled estimation of the heat–hospitalization association at the national level and within different socioeconomic levels. Based on 1,814 city-specific effect estimates, we used random effect meta-regression with each socioeconomic variable (including literacy rate, urbanization rate, household income, and log-transformed GDP per capita) as the only meta-predictor to test whether the city-level heat–hospitalization association was significantly modified by each city-level socioeconomic variable. To capture any potential non-linearity in the meta-regression, we tried to add each socioeconomic variable to the meta-regression model as a natural cubic spline function with 2 to 4 degrees of freedom. However, the Bayesian information criterion (BIC) values of nonlinear models were generally larger than or close to the BIC value of the linear meta-regression model (Table B in S2 Text). This suggested that the linear model outperformed the nonlinear models in the meta-regression; thus, we used the linear meta-regression model throughout the analyses. We stratified all the analyses above by sex, 4 age groups (0–19 years, 20–39 years, 40–59 years, 60 years or above), and 16 specific causes of hospitalization.

The heat–hospitalization association was reported as the cumulative relative risk (RR) of hospitalization (with 95% CIs) over lag 0–7 days associated with every 5°C increase in daily mean temperature during the hot season. The increased hospitalization risk associated with every 5°C increase in daily mean temperature was then calculated as $100\% \times (\text{RR} - 1)$.

We performed 2 sensitivity analyses to test the robustness of our results. First, we tested whether the socioeconomic disparities in heat vulnerability remained significant after adjusting for potential confounders in the meta-regression. For each socioeconomic indicator in the random effect meta-regression model, we adjusted for city-specific mean temperature in the hot seasons, temperature range (maximum temperature minus minimum temperature in the hot seasons), and the ratio of young population (0–19 years) and elderly population (60 years or above), as an indicator of population structure. Second, we repeated the above analyses based on 1,723 cities with relatively small population sizes (ranging from 8,014 to 243,270), after excluding cities with population sizes larger than the 95th percentile of all 1,814 cities. Compared to large cities, the city-level socioeconomic indicators in those small cities could be more representative of the residents' true socioeconomic situations.

All data analyses were performed with R software (version 3.5.1). The "dlnm" and "mvmeta" packages were used for the first-stage and second-stage analyses, respectively [28,29].

## Results

A total of 49,145,997 hospital admissions (58.5% female) from 1,814 Brazilian cities in the hot seasons between 2000 and 2015 were included in the analyses. The median age of the included hospitalized patients was 33.3 years, with an IQR of 19.8–55.7 years. The case number and age distribution of hospitalizations due to 16 specific causes are reported in Table A in S2 Text. Briefly, hospitalizations due to cardiovascular diseases, COPD, and diabetes were dominated by elderly people, while hospitalizations for pneumonia, asthma, and heat illness were dominated by children and adolescents. The daily mean temperature had a median value of 25.7°C (IQR: 23.9–27.5°C; minimum–maximum: 11.4–33.1°C) in the hot seasons during 2000–2015 in the 1,814 cities, ranging from 23.9°C (IQR: 22.1–25.5°C) in the southeast to 28.0°C (IQR: 27.0–28.8°C) in the northeast. The climatic, demographic, and socioeconomic indicators were correlated with each other (Table C in S2 Text). The northern cities were hotter than the southern cities, while the latter generally had a higher socioeconomic level (higher literacy rate, urbanization rate, household income, or GDP per capita) and older population structure than the former. There were huge socioeconomic variations across the 1,814 cities. For example, the 16-year average GDP per capita varied from US$978 to US$83,307 (Table 1; Fig 1; Fig B in S2 Text).

At the national level, we estimated that every 5°C increase in daily mean temperature in the hot season was associated with a 4.0% (95% CI 3.7%–4.3%; RR = 1.040, 95% CI 1.037–1.043, $P < 0.001$) increase in all-cause hospitalization. This association showed a clear strengthening with decreases in literacy rate, urbanization rate, average household income, and GDP per capita (Fig C in S2 Text). The RR (95% CI) for cities of LMI, UPI, and HI according to the World Bank's classification was 1.051 (1.044–1.057, $P < 0.001$), 1.037 (1.033–1.040, $P < 0.001$), and 1.026 (1.017–1.034, $P < 0.001$) (meta-regression $P$ value $< 0.001$), respectively (Fig 2).

The inter-city socioeconomic disparities in heat–hospitalization association were consistent between females and males, but especially significant in the young age group (0–19 years) (increased hospitalization risk [95% CI]: 9.9% [8.7% to 11.1%], $P < 0.001$, in LMI cities versus 5.2% [4.1% to 6.3%], $P < 0.001$, in HI cities; meta-regression $P$ value $<0.001$) (Fig 3; Table D in S2 Text).

At a national level, with every 5°C increase in daily mean temperature during the hot season, the increased cause-specific hospitalization risk was 30.3% (95% CI 24.4%–36.6, $P < 0.001$) for heat illness, 8.5% (95% CI 5.9%–11.1%, $P < 0.001$) for mental health conditions,

**Table 1. Socioeconomic, demographic, and climatic characteristics of 1,814 included cities in Brazil and the hospitalization cases during the hot season, 2000–2015.**

| Characteristic | Region | | | | | National |
|---|---|---|---|---|---|---|
| | North | Northeast | Central west | Southeast | South | |
| Number of cities | 28 | 662 | 128 | 622 | 374 | 1,814 |
| Population coverage (%) | 26.3 | 78.0 | 80.7 | 87.0 | 83.2 | 78.4 |
| **City characteristics** | | | | | | |
| Literacy rate of people aged 15 or above (%)* | 88.1 (83.8–90.8) | 72.9 (68.2–78.0) | 89.0 (87.3–91.1) | 92.5 (90.0–94.3) | 93.5 (90.7–95.5) | 89.1 (75.5–93.3) |
| Urbanization rate (%)* | 70.5 (62.7–86.7) | 60.6 (47.7–75.8) | 84.9 (78.1–91.6) | 89.7 (78.7–95.9) | 83.7 (70.5–92.0) | 80.8 (60.7–91.5) |
| Average monthly household income per capita (USD)* | 183 (143–250) | 112 (98–135) | 245 (218–273) | 258 (216–298) | 271 (232–318) | 217 (123–275) |
| GDP per capita (USD)* | 4,077 (3,063–5,309) | 1,982 (1,594–2,748) | 5,551 (3,880–7,518) | 5,739 (4,028–8,034) | 6,602 (4,989–8,709) | 4,406 (2,205–6,878) |
| Percentage of population aged 0–19 years* | 39.9 (36.2–46.8) | 37.7 (35.6–40.3) | 33.6 (31.8–36.3) | 30.6 (28.7–32.9) | 31.0 (29.2–33.4) | 33.4 (30.2–37.5) |
| Percentage of population aged 60 years or above* | 6.9 (6.3–7.7) | 10.7 (9.3–12.3) | 9.4 (7.7–11.3) | 12.5 (10.7–14.0) | 12.1 (10.3–13.9) | 11.5 (9.7–13.3) |
| Daily mean temperature in the hot seasons (˚C)* | 26.5 (25.1–27.9) | 28.0 (27.0–28.8) | 27.4 (26.1–28.8) | 23.9 (22.1–25.5) | 24.6 (23.2–25.9) | 25.7 (23.9–27.5) |
| Daily mean temperature in the hot seasons (˚C), minimum–maximum | 18.2–33.3 | 18.1–33.6 | 14.3–33.5 | 14.2–32.1 | 9.2–31.8 | 11.4–33.1 |
| **Hospitalization characteristics** | | | | | | |
| Number of cases | 1,271,435 | 13,823,251 | 3,847,427 | 22,077,029 | 8,126,855 | 49,145,997 |
| Female, n (%) | 815,026 (64.1) | 8,647,071 (62.6) | 2,281,986 (59.3) | 12,563,961 (56.9) | 4,604,677 (56.7) | 28,912,721 (58.8) |
| **Age, n (%)** | | | | | | |
| 0–19 years | 426,200 (33.5) | 4,123,994 (29.8) | 1,032,481 (26.8) | 4,860,233 (22.0) | 1,847,815 (22.7) | 12,290,723 (25.0) |
| 20–39 years | 512,158 (40.3) | 4,996,318 (36.1) | 1,351,131 (35.1) | 7,125,403 (32.3) | 2,472,683 (30.4) | 16,457,693 (33.5) |
| 40–59 years | 175,960 (13.8) | 2,337,994 (16.9) | 763,057 (19.8) | 5,152,763 (23.3) | 1,880,986 (23.1) | 10,310,760 (21.0) |
| ≥60 years | 146,161 (11.5) | 2,199,275 (15.9) | 651,016 (16.9) | 4,584,110 (20.8) | 1,787,342 (22.0) | 9,367,904 (19.1) |

For daily mean temperature, the median (IQR), minimum, and maximum were based on all city-specific daily observations for the 1,814 cities in the hot seasons between 2000 and 2015. Both GDP per capita and household income have been adjusted to 2015 USD according the Consumer Price Index. The literacy rate, urbanization rate, and GDP per capita were the 16-year average values during 2000–2015. Household income and population structure were sourced from the 2010 census of Brazil. The hot season was defined as the city-specific 4 adjacent hottest months each year. The minimum and maximum temperature were represented by the 0.1% and 99.9% quantile of daily mean temperature during the study period, in order to minimize the effects of some extreme observations.

*Median (IQR) of 1,814 cities.

GDP, gross domestic product; IQR, interquartile range; USD, United States dollars.

7.2% (95% CI 1.5%–13.1%, $P = 0.012$) for heart rhythm disturbances, 6.2% (95% CI 4.8%–7.5%, $P < 0.001$) for renal diseases, 5.2% (95% CI 4.1%–6.2%, $P < 0.001$) for pneumonia, 4.6% (95% CI 2.7%–6.5%, $P < 0.001$) for diabetes, 4.0% (95% CI 2.3%–5.7%, $P < 0.001$) for COPD, 3.0% (95% CI 1.5%–4.5%, $P < 0.001$) for peripheral vascular disease, and 3.0% (95% CI 2.2%–3.7%, $P < 0.001$) for all respiratory diseases as a whole. The same temperature rise was associated with a reduction in hospitalizations for heart failure (−3.2%, 95% CI −4.5% to −2.0%, $P < 0.001$), cerebrovascular diseases (−1.6%, 95% CI −3.0% to −0.2%, $P = 0.028$), and stroke (−1.9%, 95% CI −3.4 to −0.4, $P = 0.013$), and cardiovascular diseases as a whole (−1.1%, 95% CI −1.8% to −0.5%, $P < 0.001$). Non-significant heat–hospitalization associations were found for hospitalizations due to neoplasms, asthma, and ischemic heart disease (Fig 4; Table D in S2 Text).

The inter-city socioeconomic disparities in heat–hospitalization association were evident for hospitalizations due to certain diseases, including ischemic heart disease (increase in

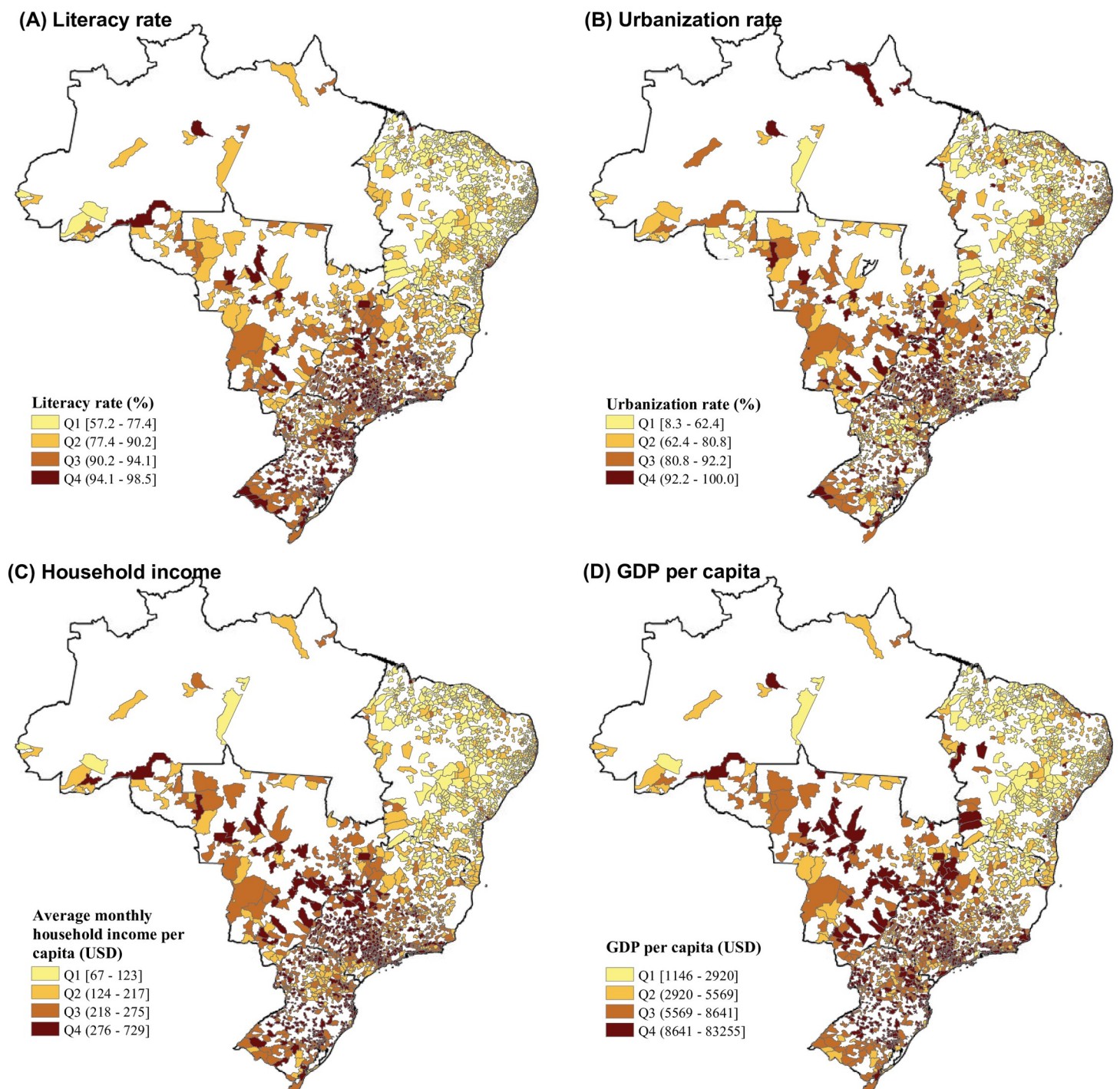

**Fig 1. Mapping the average socioeconomic characteristics during 2000–2015 in 1,814 Brazilian cities.** Literacy rate (A), urbanization rate (B), household income (C), and GDP per capita (D). Both GDP per capita and household income were adjusted to 2015 USD according to the Consumer Price Index. The literacy rate, urbanization rate, and GDP per capita were the 16-year average values during 2000–2015. Household income was sourced from the 2010 census of Brazil. Q1–Q4 were the 4 quartiles from the lowest to the highest. The base map of this figure was downloaded from the Brazilian Institute of Geography and Statistics (https://www.ibge.gov.br/); the base map was free and open-access. GDP, gross domestic product; USD, United States dollars.

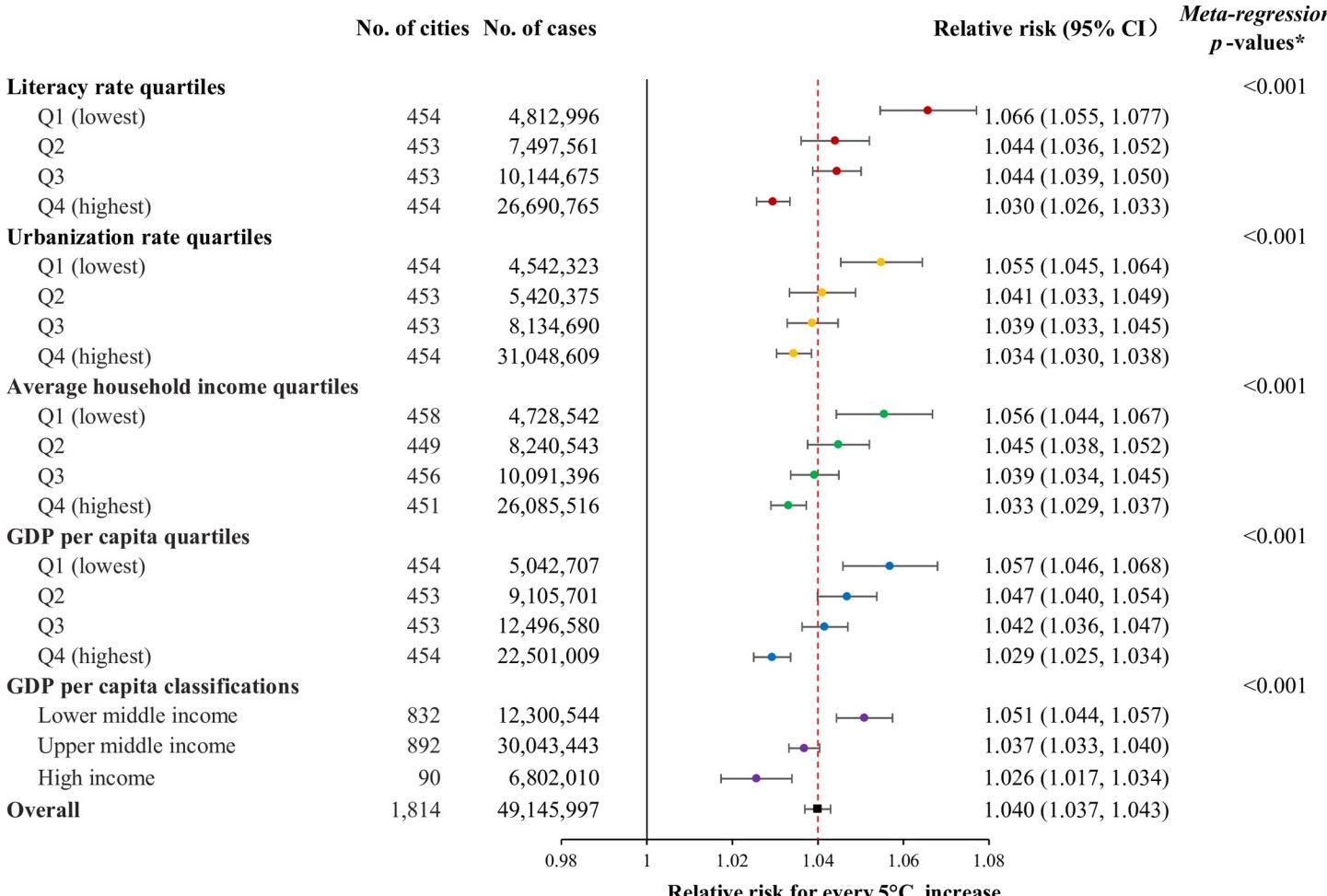

| | No. of cities | No. of cases | Relative risk (95% CI) | Meta-regression p-values* |
|---|---|---|---|---|
| **Literacy rate quartiles** | | | | <0.001 |
| Q1 (lowest) | 454 | 4,812,996 | 1.066 (1.055, 1.077) | |
| Q2 | 453 | 7,497,561 | 1.044 (1.036, 1.052) | |
| Q3 | 453 | 10,144,675 | 1.044 (1.039, 1.050) | |
| Q4 (highest) | 454 | 26,690,765 | 1.030 (1.026, 1.033) | |
| **Urbanization rate quartiles** | | | | <0.001 |
| Q1 (lowest) | 454 | 4,542,323 | 1.055 (1.045, 1.064) | |
| Q2 | 453 | 5,420,375 | 1.041 (1.033, 1.049) | |
| Q3 | 453 | 8,134,690 | 1.039 (1.033, 1.045) | |
| Q4 (highest) | 454 | 31,048,609 | 1.034 (1.030, 1.038) | |
| **Average household income quartiles** | | | | <0.001 |
| Q1 (lowest) | 458 | 4,728,542 | 1.056 (1.044, 1.067) | |
| Q2 | 449 | 8,240,543 | 1.045 (1.038, 1.052) | |
| Q3 | 456 | 10,091,396 | 1.039 (1.034, 1.045) | |
| Q4 (highest) | 451 | 26,085,516 | 1.033 (1.029, 1.037) | |
| **GDP per capita quartiles** | | | | <0.001 |
| Q1 (lowest) | 454 | 5,042,707 | 1.057 (1.046, 1.068) | |
| Q2 | 453 | 9,105,701 | 1.047 (1.040, 1.054) | |
| Q3 | 453 | 12,496,580 | 1.042 (1.036, 1.047) | |
| Q4 (highest) | 454 | 22,501,009 | 1.029 (1.025, 1.034) | |
| **GDP per capita classifications** | | | | <0.001 |
| Lower middle income | 832 | 12,300,544 | 1.051 (1.044, 1.057) | |
| Upper middle income | 892 | 30,043,443 | 1.037 (1.033, 1.040) | |
| High income | 90 | 6,802,010 | 1.026 (1.017, 1.034) | |
| **Overall** | 1,814 | 49,145,997 | 1.040 (1.037, 1.043) | |

Relative risk for every 5°C increase

**Fig 2. The association between heat exposure (for every 5°C increase in daily mean temperature) and all-cause hospitalization during lag 0–7 days, stratified by socioeconomic level.** Q1–Q4 represent 4 quartiles from the lowest to the highest. GDP per capita classifications were based on the World Bank's 2015 standard. *The meta-regression P values were derived from the meta-regressions modeling the 1,814 city-specific effect estimates against the city-level literacy rate, urbanization rate, average household income, and log(GDP per capita) separately (see Fig C in S2 Text). CI, confidence interval; GDP, gross domestic product.

cause-specific hospitalization risk with every 5°C increase [95% CI]: 5.6% [−0.2% to 11.8%], $P = 0.060$, in LMI cities versus 0.5% [−2.1%–3.1%], $P = 0.717$, in HI cities; meta-regression $P$ value = 0.029), asthma (3.7% [0.3%–7.1%], $P = 0.031$, versus −6.4% [−12.1% to −0.3%], $P = 0.041$; meta-regression $P$ value < 0.001), pneumonia (8.0% [5.6%–10.4%], $P < 0.001$, versus 3.8% [1.1%–6.5%], $P = 0.005$; meta-regression $P$ value = 0.001), renal diseases (9.6% [6.2%–13.1%], $P < 0.001$, versus 4.9% [1.8%–8.0%], $P = 0.002$; meta-regression $P$ value = 0.073), mental health conditions (17.2% [8.4%–26.8%], $P < 0.001$, versus 5.5% [−1.4% to 13.0%], $P = 0.121$; meta-regression $P$ value = 0.034), and neoplasms (3.1% [0.7%–5.5%], $P = 0.011$, versus −0.1% [−2.1% to 2.0%], $P = 0.939$; meta-regression $P$ value = 0.001) (Fig 4; Table D in S2 Text). The inter-city socioeconomic disparity patterns by sex, age, and specific cause of hospitalization were generally similar when stratifying the analyses by other socioeconomic indicators (literacy rate, urbanization rate, or household income) (Figs D–F and Tables E–H in S2 Text).

After adjusting for city-level mean temperature, temperature range, and population structure in meta-regression, the city-specific RRs were still negatively associated with literacy rate

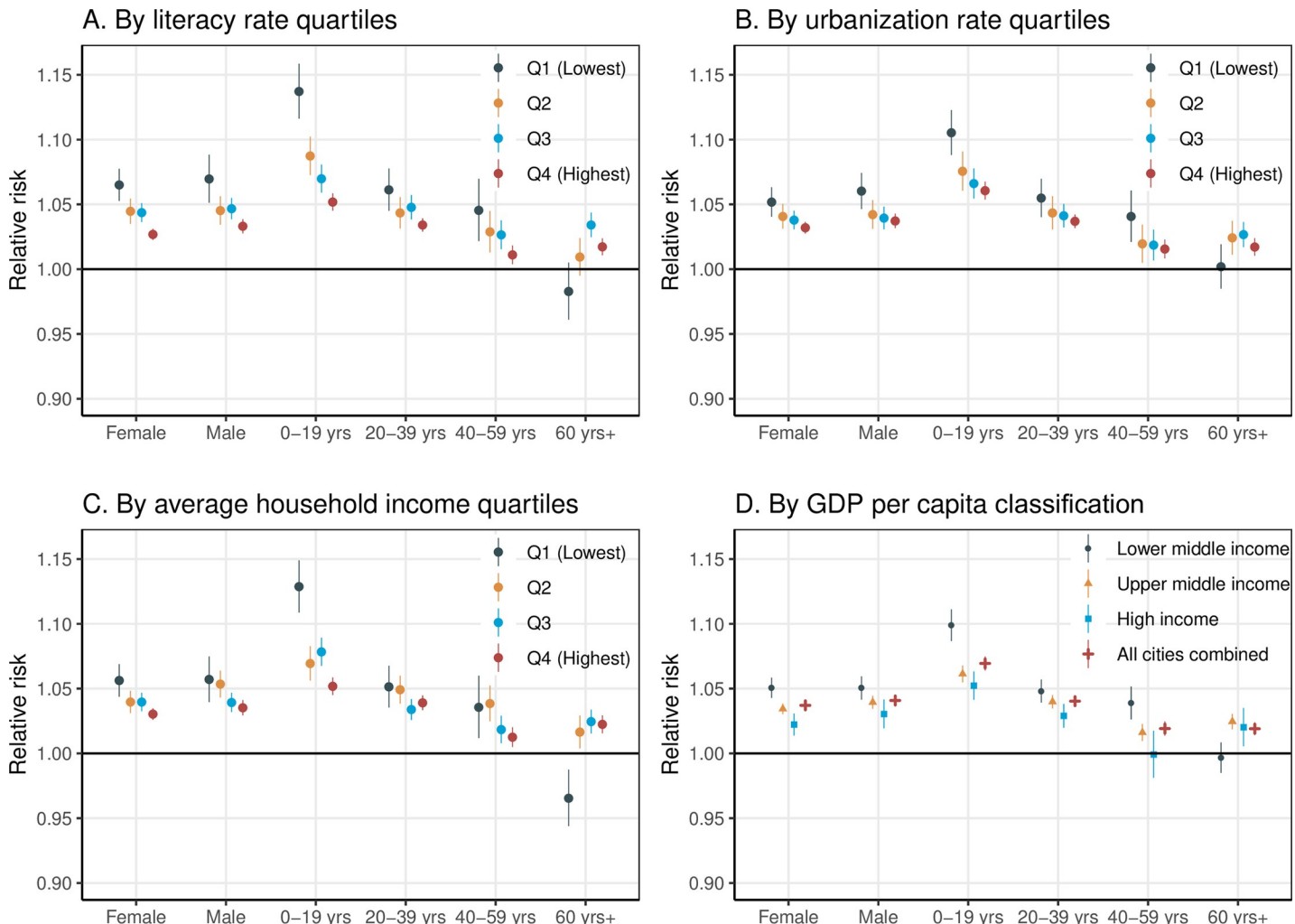

**Fig 3. The association between heat exposure (for every 5°C increase in daily mean temperature) and hospitalization during lag 0–7 days, stratified by socioeconomic level, sex, and age group.** Literacy rate (A), urbanization rate (B), household income (C), and GDP per capita (D). Q1–Q4 represent 4 quartiles from the lowest to the highest. GDP per capita classifications were based on the World Bank's 2015 standard. CI, confidence interval; GDP, gross domestic product.

(meta-regression *P* value = 0.024), urbanization rate (meta-regression *P* value = 0.033), average family income (meta-regression *P* value = 0.015), and log(GDP per capita) (meta-regression *P* value = 0.004) (Fig 5). The modifying effects of the 4 socioeconomic indicators on city-specific heat–hospitalization association changed minimally after excluding cities with population sizes larger than the 95th percentile of the 1,814 cities (Figs G and H in S2 Text).

## Discussion

This is so far the largest nationwide study to our knowledge to examine the inter-city socioeconomic disparity in the vulnerability to heat-related all-cause and cause-specific hospitalization. We found that cities with lower literacy rates, urbanization rates, average household incomes, or GDP per capita displayed stronger heat–hospitalization associations. This inter-city socioeconomic inequality was especially significant for hospitalizations of young people (0–19 years), and for hospitalizations due to ischemic heart disease, asthma, pneumonia, renal diseases, mental health conditions, and neoplasms.

## By GDP per capita classification

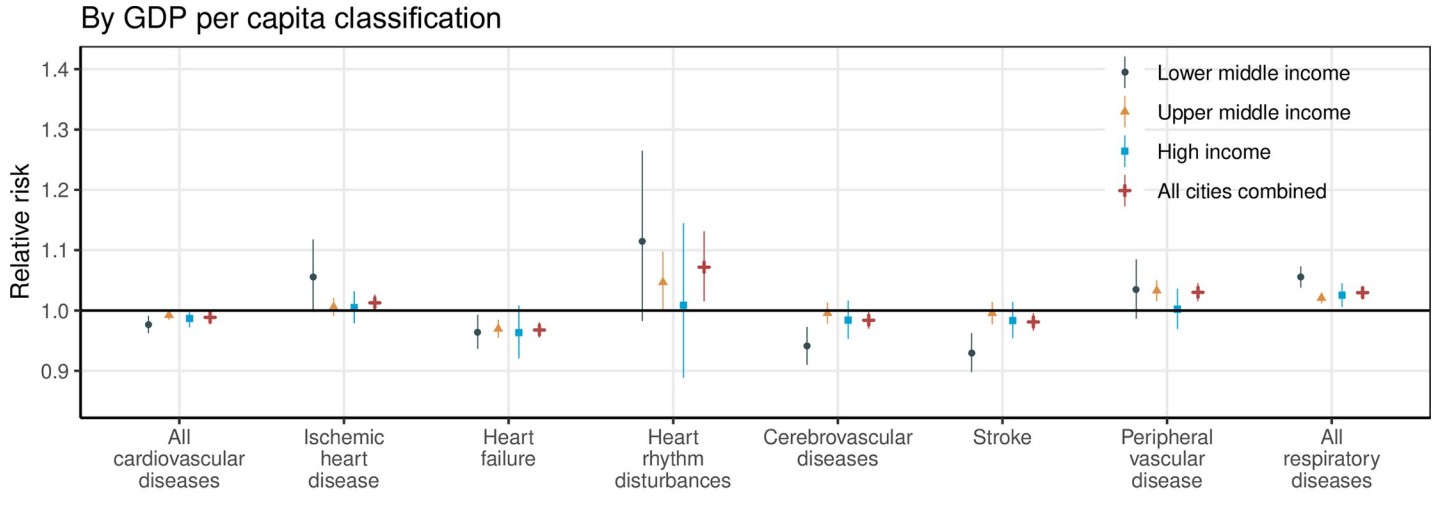

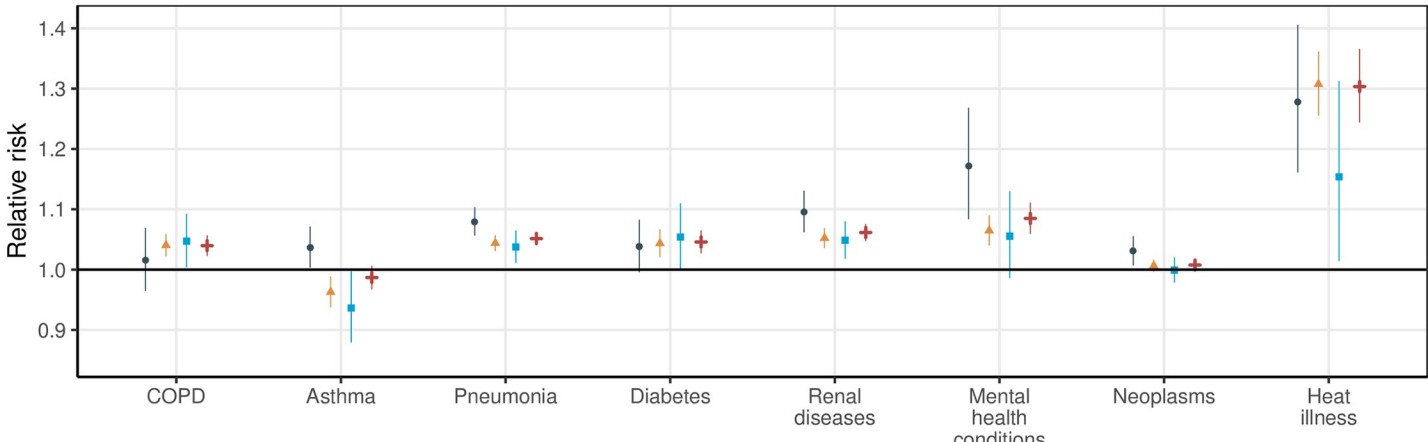

**Fig 4. The association between heat exposure (for every 5˚C increase in daily mean temperature) and cause-specific hospitalization during lag 0–7 days, stratified by GDP per capita classification.** GDP per capita classifications were based on the World Bank's 2015 standard. COPD, chronic obstructive pulmonary disease; GDP, gross domestic product.

The inter-city socioeconomic inequality in the vulnerability to heat could be explained by many factors under a framework of 3 aspects: heat exposure, sensitivity, and adaptation capacity [30]. Regarding heat exposure, people living in areas with low SES are not likely to work in climate-controlled settings (e.g., with air conditioning) [31]. They tend to perform outdoor work such as farming, construction, and mining [4,6]. Regarding sensitivity, people of low SES in Brazil are disproportionally affected by pre-existing non-communicable diseases such as heart disease, depression, chronic kidney diseases, and asthma [32], which could increase their vulnerability to heat [30]. The relatively poor sanitation conditions in areas with low SES might contribute to the residents' increased vulnerability to heat-related infectious diseases (e.g., pneumonia). Regarding adaptation capacity, people of low SES usually lack the budget to buy air conditioning [33], while the cheaper electric fans are not recommended for dealing with hot weather due to the potential risk of increasing dehydration [34]. The most recent available data for household adoption rate of air conditioning at the city level was from the 2000 Brazil census, and the adoption rate was highest for wealthy, urban households in warm

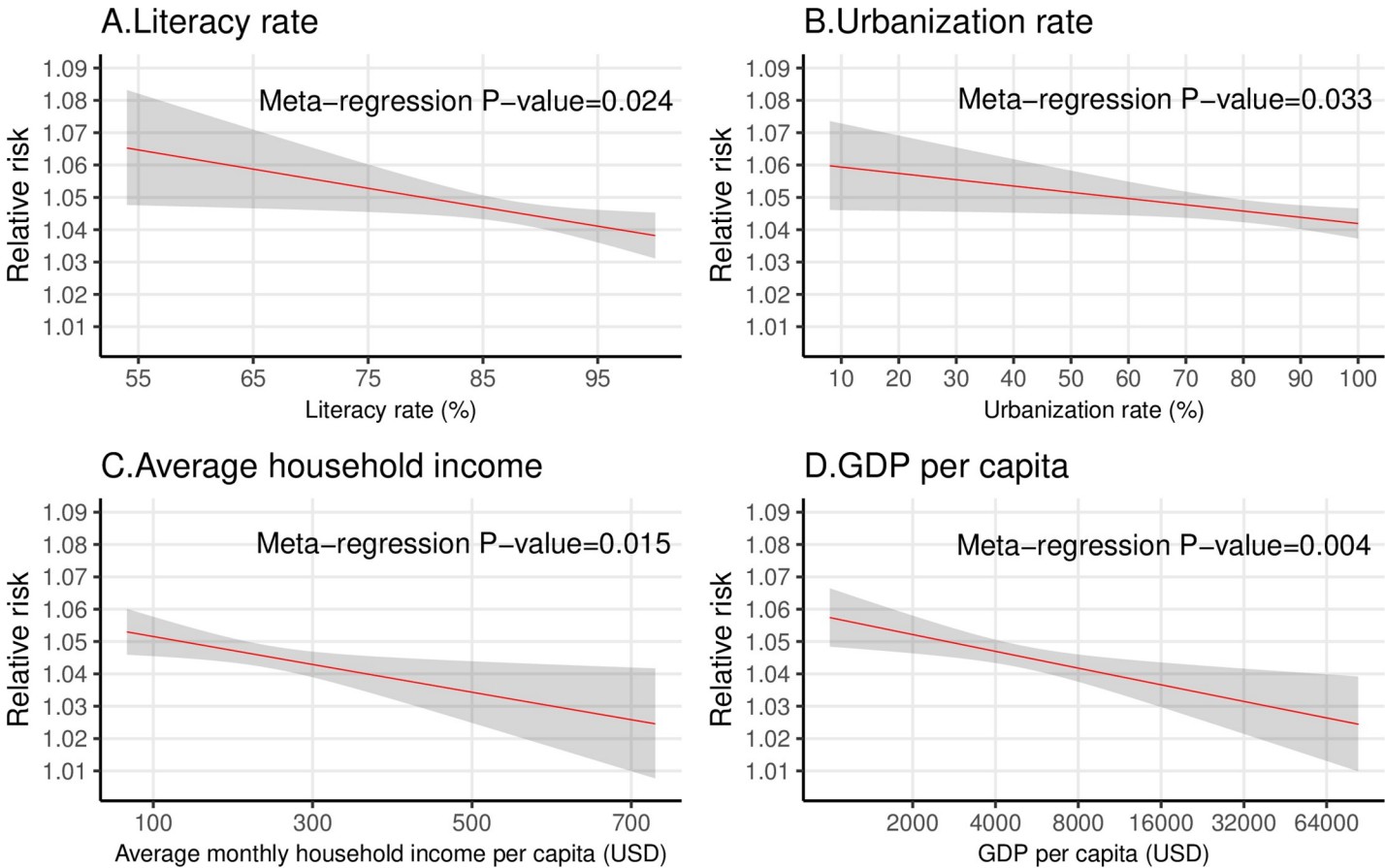

**Fig 5. The relationship between city-level socioeconomic factors and the magnitude of heat–hospitalization association among 1,814 cities, adjusting for potential confounders.** Literacy rate (A), urbanization rate (B), household income (C), and GDP per capita (D). The RR represents the association between heat exposure (for every 5°C increase in daily mean temperature) and hospitalization during lag 0–7 days. The shadowed area represents the 95% confidence interval. The relationships between RR and the 4 socioeconomic indicators were fitted separately by meta-regression, adjusting for city-specific mean temperature, temperature range, and the ratio of elderly population (≥60 years) and young population (0–19 years). The RRs were estimated as the values when city-specific mean temperature, temperature range, and the ratio of elderly population and young population were at the average level of the 1,814 cities. The x-axis of (D) is on the log scale, because we added log(GDP per capita) rather than GDP per capita to the meta-regression model. GDP, gross domestic product; RR, relative risk; USD, United States dollars.

cities [35]. In 2010, Brazil's national adoption rate of air conditioning was estimated to be 11.8% (up from 7.4% in 2000), with the lowest adoption rate (6.1%) seen in the northeast region (the hottest, poorest, and least urbanized region in Brazil) [35]. People with low educational levels may lack basic knowledge about heat-related health risks and how to prevent the risks [4,31]. In addition, less developed cities might lack cool public spaces such as shaded parks, or libraries, shopping centers, cinemas, or museums with air conditioning [31,36]. These places serve as important cooling shelters for residents during hot weather.

Our results were consistent with a previous study in Australia that included 158 areas within Brisbane and found that the heat–hospitalization associations attenuated in HI areas [18]. By contrast, a study in 132 districts of Vietnam found that the heat–hospitalization associations were stronger in more urbanized areas, which was likely to be explained by the UHI effect [19]. Our inconsistency with this Vietnamese study suggests that the impacts of urbanization on heat vulnerability could be 2-fold. On the one hand, urbanization is associated with UHI effects, which could increase urban residents' magnitude and duration of heat exposure [37,38]. This could exacerbate the risk of heat-related mortality and morbidity among urban

residents [10,11,19]. On the other hand, urbanization is also accompanied by socioeconomic improvements, such as improvements in education, income, access to air conditioning, housing, and sanitation conditions, and decreases in outdoor working times and pre-existing health conditions [4,6,31,32,35]. All these factors tend to reduce urban residents' sensitivity and increase their adaptation capacity in response to heat exposure. This could explain the attenuated heat–hospitalization or heat–mortality associations in highly urbanized areas observed in our study and previous studies [3,4].

Therefore, whether urbanization is associated with increased or decreased heat vulnerability may depend on the balance between UHI effects and socioeconomic improvements. In this study, for highly urbanized cities of Brazil, the potential effects of UHI effects on heat vulnerability was likely to be offset by the socioeconomic improvements associated with urbanization. Our study and 2 previous studies in China [3,4] challenge the idea of treating high urbanization as an index of high heat vulnerability, as proposed in a recent report [1]. Without considering socioeconomic improvements along with urbanization, it would be inappropriate to conclude that high urbanization would definitely increase heat vulnerability because of UHI effects [1]. More weight should be given to socioeconomic indicators when constructing a heat vulnerability index to monitor the health impacts of heat exposure.

Although previous studies have evaluated the potential modification effect of local SES on heat–mortality [3–11,13–16] or heat–morbidity association [17–20], few of them stratified the analyses by sex, age, and specific causes of hospitalization. The similar socioeconomic disparity in heat vulnerability for females and males is consistent with 2 heat–mortality studies [4,16]. The more significant modification effect found in younger (0–19 years) than other ages could well be explained by the higher heat–hospitalization association in this age group, possibly due to the high prevalence of pneumonia and asthma compared to other ages [23]. For hospitalizations due to asthma, the finding that heat exposure was a risk factor in LMI cities while a protective factor in UMI and HI cities was not totally unexpected. A recent review also found such a discrepancy between studies from different cities [39]. Temperature rise within a certain range could have both adverse (through increasing indoor aeroallergens like spores and cockroaches) and beneficial (through enhancing lung capacity and immune system) impacts on asthma morbidity [39]. In less developed cities with poor sanitation conditions, the adverse impacts might outweigh the beneficial impacts.

The protective effect of heat exposure for hospitalization due to heart failure has also been reported by a US study [40]. It is possible that heart volume overload becomes less likely due to fluid loss in hot weather [40]. The protective effect of heat exposure for hospitalization due to stroke in LMI cities is likely to be due to a decreased rate of hemorrhagic stroke (see Fig I in S2 Text). A meta-analysis also found that high temperatures could reduce the risk of hemorrhagic stroke, which might be explained by the reduced blood pressure due to fluid loss and dilated peripheral vessels in hot temperatures [41]. Most hospitalizations due to cerebrovascular diseases (mainly stroke) and heart failure happened in elderly people (Table A in S2 Text), especially those living in LMI cities (Table I in S2 Text). This might explain the unexpected inter-city socioeconomic disparity pattern in heat vulnerability for elderly people.

The public health implications of this study are 2-fold. On the one hand, given the stronger heat–hospitalization associations in cities of lower SES, climate change could be a potential driver to exacerbate the inter-city inequality in Brazil. Less developed cities in Brazil are expected to suffer more from global warming and heat-related healthcare burden and its related economic costs [42], although they contribute less to carbon emissions than more developed cities. To tackle this unfair situation, more resources (e.g., heat warning systems, air conditioning) should be invested in less developed cities in Brazil under a changing climate. On the other hand, our study might also suggest a potential declining heat–hospitalization

association along with economic growth in Brazil. In other words, there might be adaptation to heat exposure as reported in the US, Czech Republic, Spain, and Japan [43,44]. However, some other countries (e.g., Australia and South Korea [43]) did not show significant adaptation to heat. Our previous study reported a non-significant decline of heat–hospitalization association during 2000–2015 in Brazil, despite rapid economic growth during this period [23]. Therefore, more studies are needed to explore whether socioeconomic improvement could promote adaptations to heat.

The present study has several strengths. First, the 16-year-long study period, very large sample size (49 million hospitalizations), and large number of locations (1,814 cities) ensured the statistical power and robustness of our results. Second, to our knowledge, ours is the first study evaluating almost all major cause-specific heat-related hospitalizations in one study. This study could give people a comprehensive view on heat–morbidity associations. Going beyond mortality to morbidity statistics (represented by hospitalization) also adds great value to existing literature, which mainly focuses on heat–mortality associations. Third, the national hospitalization dataset covered nearly 80% of the Brazilian population, which gives good representativeness. Finally, given the great socioeconomic diversity within the 1,814 cities studied, our findings might also apply to other middle- and high-income countries.

There are also several limitations. Some limitations have been discussed by our previous publication, including the use of gridded temperature data rather than personal measurements, and being unable to adjust for relative humidity [23]. Due to the unavailability of individual-level socioeconomic data in the hospitalization records, we had to rely on city-level socioeconomic data, which meant we could not evaluate intra-city or individual-level socioeconomic inequalities [30]. However, our findings could still be quite relevant for adaptation policy at the state or national level, in terms of allocating resources between cities to deal with the increasing heat exposure under global warming. This could be an indispensable part of promoting Brazil's within-country health equity, one of the priorities of the Sustainable Development Goals [45]. The 4 socioeconomic indicators in our study were correlated with each other, making it difficult to identify and compare the independent modification effect of each indicator [31]. However, the results from different indicators were largely similar, and all suggested the existence of inter-city socioeconomic disparities in heat vulnerability.

Finally, due to data availability, we were unable to adjust for some potential important city characteristics (e.g., air quality, green space, population health status, access to air conditioning) that could explain or confound the inter-city socioeconomic disparities. As discussed, access to air conditioning and population health status (e.g., prevalence of chronic diseases) tend to be mediators between city-level SES and heat vulnerability [30,32,33,35]. Adjusting for these potential mediators in future studies would help to test hypotheses about the potential pathways of socioeconomic disparity in heat vulnerability. Economic growth and urbanization in Brazil are generally associated with increased ambient air pollution and decreased green space (a measure of vegetation density), due to increased emissions from industries and traffic and a shift from vegetated lands to building structures [46–49].Poor outdoor air quality and low green space have been found to enhance rather than attenuate residents' vulnerability to heat-related morbidity and mortality [50–54]. Therefore, the negative association between city-level socioeconomic factors and heat vulnerability is not likely to be explained by inter-city variations in green space and air quality.

In conclusion, less developed cities in Brazil displayed stronger heat–hospitalization associations, especially for hospitalizations of children and adolescents, and for hospitalizations due to specific conditions. This may exacerbate existing inter-city health and socioeconomic inequalities. More resources should be invested in less developed cities in Brazil to tackle the morbidity burden of heat exposure and to promote health equity under global warming.

## Supporting information

**S1 STROBE Checklist. Checklist of items that should be included in reports of observational studies.** STROBE, Strengthening the Reporting of Observational Studies in Epidemiology.
(DOCX)

**S1 Text. Prospective analysis plan and modifications following comments from editors and reviewers.**
(DOCX)

**S2 Text. Supplementary tables and figures.**
(DOCX)

## Acknowledgments

We thank the Brazilian Ministry of Health and Brazilian National Institute of Meteorology for providing hospitalization and meteorological data, respectively.

## Author Contributions

**Conceptualization:** Rongbin Xu, Shanshan Li, Yuming Guo.

**Data curation:** Rongbin Xu, Qi Zhao, Micheline S. Z. S. Coelho, Paulo H. N. Saldiva, Shanshan Li, Yuming Guo.

**Formal analysis:** Rongbin Xu.

**Funding acquisition:** Micheline S. Z. S. Coelho, Shanshan Li, Yuming Guo.

**Investigation:** Rongbin Xu, Paulo H. N. Saldiva, Yuming Guo.

**Methodology:** Rongbin Xu, Shanshan Li.

**Project administration:** Yuming Guo.

**Resources:** Paulo H. N. Saldiva.

**Software:** Rongbin Xu.

**Supervision:** Qi Zhao, Paulo H. N. Saldiva, Michael J. Abramson, Shanshan Li, Yuming Guo.

**Validation:** Yuming Guo.

**Visualization:** Rongbin Xu.

**Writing – original draft:** Rongbin Xu.

**Writing – review & editing:** Qi Zhao, Micheline S. Z. S. Coelho, Paulo H. N. Saldiva, Michael J. Abramson, Shanshan Li, Yuming Guo.

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
