## [Editor Report · Decision Letter 0]

8 Mar 2020

Dear Dr Xu, 

Thank you for submitting your manuscript entitled "Socioeconomic disparity in heat impacts on all-cause and cause-specific hospitalisation in 1,814 Brazilian cities" for consideration by PLOS Medicine.

Your manuscript has now been evaluated by the PLOS Medicine editorial staff and I am writing to let you know that we would like to send your submission for external assessment.

However, we first need you to complete your submission by providing the metadata that is required for full assessment. To this end, please login to Editorial Manager where you will find the paper in the 'Submissions Needing Revisions' folder on your homepage. Please click 'Revise Submission' from the Action Links and complete all additional questions in the submission questionnaire.

Kind regards,

Richard Turner PhD, for Clare Stone, PhD,

Senior editor, PLOS Medicine

rturner@plos.org

---

## [Decision Letter · Decision Letter 1]

25 Jun 2020

Dear Dr. Xu,

Thank you very much for submitting your manuscript "Socioeconomic disparity in heat impacts on all-cause and cause-specific hospitalisation in 1,814 Brazilian cities: a nationwide case-crossover study" (PMEDICINE-D-20-00704R1) for consideration at PLOS Medicine. 

[LINK]

In light of these reviews, I am afraid that we will not be able to accept the manuscript for publication in the journal in its current form, but we would like to consider a revised version that addresses the reviewers' and editors' comments. Obviously we cannot make any decision about publication until we have seen the revised manuscript and your response, and we plan to seek re-review by one or more of the reviewers. 

We expect to receive your revised manuscript by Jul 16 2020 11:59PM. Please email us (plosmedicine@plos.org) if you have any questions or concerns.

We look forward to receiving your revised manuscript. 

Sincerely,

Emma Veitch, PhD

PLOS Medicine

On behalf of Clare Stone, PhD, Acting Chief Editor,

PLOS Medicine

plosmedicine.org

*Please structure the abstract using the PLOS Medicine headings (Background, Methods and Findings, Conclusions - "Methods and Findings" should be a single subsection heading). 

*In the last sentence of the Abstract Methods and Findings section, please describe the main limitation(s) of the study's methodology.

*At this stage, we ask that you include a short, non-technical Author Summary of your research to make findings accessible to a wide audience that includes both scientists and non-scientists. The Author Summary should immediately follow the Abstract in your revised manuscript. This text is subject to editorial change and should be distinct from the scientific abstract. Please see our author guidelines for more information: https://journals.plos.org/plosmedicine/s/revising-your-manuscript#loc-author-summary

*Please clarify if the analytical approach used in this paper corresponded to one set out in a prospectively-developed protocol or analysis plan? Please state this (either way) early in the Methods section.

*It's notable that the authors acknowledge as a limitation, and the reviewers note, that associations reported in the paper correspond to an ecological analysis, ie at the level of city, not at the level of individual. However, as noted by reviewers, the writeup in the paper doesn't always take this into account and at times slips into claiming effects at the level of the individual - more care should be taken around this. In addition, it would be good to more explicitly state in the abstract the ecological analysis ie that associations are at the level of the city. 

*Currently the limitations section of the Discussion is quite brief and this doesn't say anything about the potential for effects to be explained by confounding due to unmeasured variables, and this could also be considered/discussed. 

Comments from the reviewers:

Reviewer #1: I confine my remarks to statistical aspects of this paper. 

I am concerned with the whole city-by-city method. It's not that you can't do analysis on a city basis, but you cannot draw inferences about individuals from city level data - this is known as the ecological fallacy. So, for example, in the first paragraph of the discussion, you cannot say that it was worse for people of a particular age, only for cities with particular age distributions. Throughout the discussion, the authors make statements about individuals that are not justified. 

(The ecological fallacy is really counter-intuitive and easy to fall into). 

On a more mundane level, the authors should not categorize the independent variables. In *Regression Modeling Strategies* Frank Harrell lists 11 problems with doing this and sums up "nothing could be more disastrous". Leave the various rates as continuous variables and use a spline to investigate nonlinearity. (It is OK to graph quartiles, but they should not be used for modeling.)

Peter Flom

Reviewer #2: This is a well written paper that should be an important contribution to the literature. I have a few comments that could improve this paper.

A couple of broadly applicable comments are as follows; with examples; additional line by line comments appear later.

The paper is overall very clearly written. However, some minor grammatical issues appear throughout that suggest a review by a native English speaker could be useful.

For example, in the abstract, page 2, lines 23 and 24, the word 'the' is not needed before the word 'vulnerability'.

Second, these authors at times use the phrase 'due to' (page 2, line 39) which implies causality. I suggest replacing 'due to' in this instance with 'associated with'; the same substitution is recommended in other places.

Additional comments are here:

Page 2, lines 39-42; why no confidence intervals?

Page 2, line 43, disparity should be plural, and stratified should be 'stratifying'

Page 3, line 46, here again, suggest replacing due to with 'associated with'.

Page 3, line 47, spell out GDP at first use.

Page 5 lines 93-4. Were secondary diagnoses available and if so why were they not used in classification? Please explain this. 

Page 5 line 95 why is the term 'also' included?

Page 5 line 101: were exemptions also granted for Brazilian authors or did they not have access to the data? Presumably someone had to process the data with individual identifiers before it was ready for analysis.

Page 6 line 198; it would be clearer to say '16-year-average' as age is described just previously.

Page 7 line 7, should say 'remained', not 'retained.'

Page 18, lines 158- 163. I can understand why authors may not want to evaluate the city characteristics they describe; (air quality, vegetation, land cover, built environment, health status), but I don't find the reasoning at all convincing. On what basis are they expected to be 'affected by city level SES rather than the converse?' Why would this be the case in Brazil, by what mechanism? If data on these characteristics are available,I would think it possible to include them in the manner done in a recent related article, Gronlund CJ, Berrocal VJ, White-Newsome JL, Conlon KC, O'Neill MS. Vulnerability to extreme heat by socio-demographic characteristics and area green space among the elderly in Michigan, 1990-2007. Environ Res. 2015;136:449-61.

This section is my reason for suggestion 'major revision' because the section and then later discussion about the interplay between heat exposure and SES just seems too 'facile' and surface level. 

Even the comment later about the area-level SES variables being correlated thus hampering interpretation would be better illustrated quantitatlively.

Page 8 line 173, hospitalisitions should be plural.

Page 9 Table 1: Why not just spell out Number in lines 1 and 3?

Why not just say percent instead of proportion in the 3rd and fourth from last lines?

Page 10, line 186, correct spelling of characteristics; 

Line 188, take out the word OF (last word)

Page 11, Figure 2:

I understand why the 'urbanization' is shown from lowest to highest quartile as the 'dose response is similar for this variable as the other ones, but wouldn't the original hypothesis potentially have been that higher urbanization would be linked to greater vulnerability? Just wonder if the authors might consider presenting those variables a bit differently. (minor point)

Page 13 line 223, stratified should read 'stratifying'.

Page 15, paragraph beginning at line 260 needs revision; how do these mechanisms relate to the earlier comment about SES and land cover, built environment, etc?

In addition, where are the data on Brazil-specific prevalence of and access to air conditioning? All the citations are to large multi city studies. What is known about this and how much AC access and use differ by location, SES, etc. ? This is an omission and needs to be rectified. There have to be some data that the Brazilian coauthors could help locate, or if not, talk about the limitations and what is known.

Page 15, line 277: the comment on urbanization and how those results differ from results in Vietnam deserves further expansion.

Page 16, lines 284-7. These sentences with a recommendation do not seem well supported; why would a heat exposure vulnerability index in particular relate to the contradictory findings of this article from the Lancet recommendation? Needs revision.

Page 17, line 313; should read 'driver'

Page 17, line 321, remove the phrase 'there are also'.

Page 18, lines 339-340: were the individual level data not available on the hospitalization records? It is worth describing whether this is the case or not

Reviewer #3: This is an interesting study on a very important topic, namely that of inequalities in climate impacts. However, I do have serious reservations with the design of the study:

- My first issue is rather fundamental, and is related to the scale of the study. The authors have aggregated climate as well as socio-economic indicators on a city level. For temperature this can be defended, even though there is likely to be variation in temperature within cities. However, for socio-economic levels, I don't see this as a valid approach. The variation, especially in large cities such as Sao Paolo or Rio de Janeiro, is enormous, and using one value, e.g. average household income, to define all inhabitants seems very flawed.

- Furthermore, for some indicators, only one or two data points were available, and those were interpolated to 15 years. This is a very inaccurate way of dealing with indicators that are already not very representative for a large share of city inhabitants.

- In addition, temperature is not independent of socio-economic indicators, as the authors also indicate that cities in the South are wealthier and colder. This also leads to spurious interpretation of the findings, which occurs in various places in the Results and Discussion sections

In summary, I think the study would be publishable if the focus is only on the relationship between temperature and hospitalization, which by itself is already rather innovative. But I would remove the socio-economic component completely (second stage of the analysis), which is very inaccurate.

I also have some smaller comments:

Methods:

- The authors refer on several occasions to methodological aspects that have been described in previous publications. However, this is not sufficient for someone reading this publication by itself, and there should not be a necessity to look for additional sources.

Results:

- Line 169: what does this number of hospitalisations represent? Is this only for the warmest four months, as mentioned in the methods?

- Lines 175-177: this sentence is confusing, consider revision and clarification on what the mean and median refer to (within cities, between cities; within years, between years).

- Line 192-193: 5 °C increase above what? There should be a baseline temperature.

- There is no description of the results on the relationship between temperature and cause-specific hospitalization (independent of socio-economic indicators). This would be a very interesting addition.

[LINK]

---

## [Decision Letter · Decision Letter 2]

12 Aug 2020

Dear Dr. Xu,

Thank you very much for re-submitting your manuscript "Socioeconomic disparity in heat impacts on all-cause and cause-specific hospitalisation in 1,814 Brazilian cities: a nationwide case-crossover study" (PMEDICINE-D-20-00704R2) for review by PLOS Medicine.

I have discussed the paper with my colleagues and the academic editor and it was also seen again by 3 reviewers. I am pleased to say that provided the remaining editorial and production issues are dealt with we are planning to accept the paper for publication in the journal.

[LINK]

We look forward to receiving the revised manuscript by Aug 19 2020 11:59PM. 

Sincerely,

Artur Arikainen, 

Associate Editor 

PLOS Medicine

plosmedicine.org

Requests from Editors:

1. Following discussion with the Academic Editor, we feel that reviewer #2’s comment on representativeness in smaller cities can be left for readers to make their own judgements.

2. Title: Please amend to: “Socioeconomic level and associations between heat exposure and all-cause and cause-specific hospitalization in 1,814 Brazilian cities: a nationwide case-crossover study”

3. Short Title: Please amend to: “Socioeconomic disparities in health associations of heat”

4. Financial disclosure: Please correct the last sentence to “The funding bodies did not play any role…” or replace with the recommended phrasing: “The funders had no role in study design, data collection and analysis, decision to publish, or preparation of the manuscript.”

5. Abstract:

a. Line 21: Correct to “…increase with global warming…”

b. Please quantify the results with p values.

c. Lines 45-46: Please rephrase to: “The main limitations were that our city-level analysis did not assess…”

d. Line 48 and throughout the manuscript: Please replace “Less developed” with “Lower income”

e. Please include an additional limitation at the end of the Methods and Findings subsection.

6. Author Summary:

a. Please include bullet points at the start of each sentence.

b. Lines 61, 69, and throughout the manuscript: Please replace “less developed” with “lower income”

c. Line 63: Correct to “These inter-city socioeconomic disparities in associations of heat with health outcomes were particularly notable for…”

d. Line 67: Correct to: “along with global warming”

7. Please move your citation callouts to before punctuation and remove any spaces within the square brackets, eg. line 84: “…stronger heat-mortality associations [10,11].”

8. Methods:

a. Lines 110-111: Please change to “Following feedback during peer review, we made some modifications…”

b. Lines 134 and 171: Please replace “effect” with “association” or similar.

c. Line 138: Correct to “proportion of urban population”

9. Tables and Figures: Please provide a definition for all abbreviations (eg. USD, GDP) in the footnotes, where appropriate.

10. Results: Please quantify the results with p values in the text, eg. line 238.

11. Discussion:

a. Line 315: Please clarify that “This is so far the largest nationwide study to our knowledge…”; similarly at line 409.

b. Line 363: Please change to simply “…recent report [1].“

c. Line 407: Please replace “huge” with “very large” or similar. Also, please clarify that “49 million hospitalizations”.

d. Line 411: Replace “impacts” with “associations”.

12. Lines 453-457, and 508-522: Please remove the authors contributions, data availability, funding and competing interests – this information will be taken from the online submission form.

13. Please provide full access details (eg. URL or DOI) for references 35 and 48.

----

Comments from Reviewers:

Reviewer #1: The authors have addressed my concerns and I now recommend publication.

Peter Flom

Reviewer #2: I think the authors did a nice job responding to comments.

I noticed they added in Line 115 "According to reviewers' comments, we have made some modifications to the analysis plan in S1 Text."

I am not sure that edit belongs in the final version of the paper that will be published, of course readers understand the paper has been peer reviewed I leave it to the editors to make the decision.

Lines 216-219 "Second, we repeated the above analyses based on 1,723 cities with relatively small population sizes (ranging from 8,014 to 243,270), after excluding cities with population sizes larger than the 95th percentile of 1,814 cities. Compared to large cities, the city-level socioeconomic indicators in those small cities could be more representative of the residents' true socioeconomic situations." 

I am interested in why the assumption that the city-level indicators could be more 'representative' of individual SES was made. I personally am not convinced, there could be just as much heterogeneity in a smaller city or more…but I don't think the change was based on my comment. Again, I defer to the editors here.

Reviewer #3: The authors have responded very adequately to my previous review. I don't have further comments, except for the fact that I would refer to 'strengths' in the end of the discussion section, moreso than 'advantages' (opposite of 'limitations').

[LINK]

---

## [Editor Report · Decision Letter 3]

3 Sep 2020

Dear Mr. Xu, 

On behalf of my colleagues and the academic editor, Dr. Liz Hanna, I am delighted to inform you that your manuscript entitled "Socioeconomic level and associations between heat exposure and all-cause and cause-specific hospitalization in 1,814 Brazilian cities: a nationwide case-crossover study" (PMEDICINE-D-20-00704R3) has been accepted for publication in PLOS Medicine. 

PRODUCTION PROCESS

PRESS

PROFILE INFORMATION

Thank you again for submitting the manuscript to PLOS Medicine. We look forward to publishing it. 

Best wishes, 

Artur Arikainen, 

Associate Editor 

PLOS Medicine

plosmedicine.org